

# Phytoplankton growth and physiological responses to a plume front in the
# northern South China Sea
Qian P. Li [1,2,*], Weiwen Zhou [1,2], Yinchao Chen [1,2], and Zhengchao Wu[1]
[1]South China Sea Institute of Oceanology, Chinese Academy of Sciences, Guangzhou 510301, China
[2]University of Chinese Academy of Sciences, Beijing 100049, China
*Corresponding to*: Qian Li (qianli@scsio.ac.cn)
**Abstract.** Due to a strong river discharge during April-June 2016, a persistent salinity front, with
freshwater flushing seaward on the surface but seawater moving landward at the bottom, was formed in
the coastal waters west of the Pearl River Estuary (PRE) over the Northern South China Sea (NSCS)
shelf. Hydrographic measurements revealed that the salinity front was influenced by both river plume
and coastal upwelling. Shipboard nutrient-enrichment experiments with size-fractionation chlorophyll-*a*
measurements were performed on both sides of the front as well as the front zone to diagnose the spatial
variations of phytoplankton physiology across the frontal system. We also assessed the size-fractionated
responses of phytoplankton to the treatment of plume water at the frontal zone and the seaside of the
front. Biological impact of vertical mixing or upwelling was further examined by the response of
surface phytoplankton to the addition of local bottom water. Our results suggested that there was a large
variation of phytoplankton physiology on the seaside of the front driven by dynamic nutrient fluxes,
although P-limitation was prevailing on the shore-side of the front and at the frontal zone. The
spreading of plume water at the frontal zone would directly improve the growth of micro-phytoplankton,
while nano- and pico-phytoplankton growths could become saturated at high percentages of plume
water. Also, the mixing of bottom water would stimulate the growth of surface phytoplankton on both
sides of the front by altering the surface N/P ratio closer to the Redfield stoichiometry. In summary,
phytoplankton growth and physiology could be profoundly influenced by physical dynamics of the
frontal system during the spring-summer of 2016.



## 1 Introduction

It is well known that physical dynamics of coastal ocean can be strongly influenced by river input. When there is a high river discharge, a large plume of brackish water can form near the estuary mouth and the adjacent inner shelf regions, which is generally a low-salinity mesoscale feature that disperses fresh river water across the coastal margin (Horner-Devine et al., 2015). River plumes can transport and redistribute river-borne materials, such as nutrients and particles, and thus largely affect biogeochemistry of the coastal ocean (Dagg et al., 2004; Lohrenz et al., 2008). Convergent surface fronts over the shelf are a common feature associated with river plumes (e.g. Garvine and Monk, 1974). These plume-induced fronts are often the places of high phytoplankton productivities (Acha et al., 2004) and thus provide important feeding and reproductive habitats for higher trophic-level marine organisms, such as zooplankton and fish (Morgan et al., 2005).

Biological production of the coastal Northern South China Sea (NSCS) is controlled by monsoon-driven upwelling that brings nutrient-rich deep waters to the shelf (Liu et al., 2002). In addition, there is an intense river discharge from the Pearl River Estuary (PRE) during the spring-summer leading to the development of a strong river plume nearshore (Su, 2004). In the coastal water west of the PRE, convergence between the northeastward coastal current and the southeastward river plume can maintain a sharp salinity front along the shelf when the southwest monsoon is prevailing over the region (Wong et al., 2003). Variability of the front is primarily controlled by the river discharge and by the direction and magnitude of the regional wind field (Dong et al., 2004). On the east of the PRE, the surface plume water can be entrained in the coastal current as a salinity tongue in the summer and propelled eastward and offshore by wind-driven jets to affect the large area of the NSCS shelf-sea (Gan et al., 2009).

The plume front over the NSCS shelf creates an interface between the river plume and the adjacent marine waters with rapid changes of both salinity and nutrients at the frontal zone (e.g. Cai et al., 2004). There is a P-limitation of phytoplankton in the river plume due to a high N/P ratio of the PRE water (Zhang et al., 1999; Yin et al., 2001). In contrast, biological production is generally N-limited in the offshore oceanic waters (Wu et al., 2003; Chen et al., 2004), as the upwelled deep-water with an N/P





ratio of ~14 is essentially N-deficient compared to the Redfield N/P ratio of 16 (Wong et al., 2007). A
shift from P-limitation to N-limitation of phytoplankton community across the plume edge to the sea
has been speculated based on results of the Hong Kong waters (Yin et al., 2001). Results of a
physical-biogeochemical coupling model in the NSCS indeed predict a fast decrease of N/P ratio from
~120 in the near-field to <13.3 in the far-field of the plume front driven by a higher N/P consumption
ratio and by mixing with the ambient lower N/P water (Gan et al., 2014).
Nutrient variations, in addition to light fluctuation, can affect the partitioning of phytoplankton
biomass between different size classes (Marañón et al., 2012, 2015). The change of phytoplankton size
structure can be controlled by size-dependent trade-off processes for resource acquisition and use
(Marañón, 2015). Small phytoplankton has a higher nutrient affinity for growth under nutrient limiting
conditions (Suttle, 1991; Raven et al., 1998), whereas large phytoplankton shows higher growth
efficiency under favorable nutrient conditions (Cermeño et al., 2005). A large shift of phytoplankton
assemblage from small to large cells could arise following the addition of nutrients from deep seawater
in the North Pacific Subtropical Gyre (McAndrew et al., 2007). The success of large phytoplankton in
the oligotrophic ocean would highly depend on external environmental dynamics, although it has the
metabolic potential of enhance production (Alexander et al., 2015). It is thus important to understand
not only the mechanisms for nutrient variations, but also the response of size-fractionated
phytoplankton community to the diverse nutrient supplies, particularly at the frontal zone where the
patchiness of phytoplankton can be affected by complex physical dynamics (Li et al., 2012).
Three field surveys were carried out to study the biological response to a strong salinity front over
the NSCS shelf during the April-June of 2016. Besides comprehensive hydrographic and
biogeochemical measurements, such as temperature, salinity, nutrients, and chlorophyll-*a*, we
performed nutrient-enrichment experiments with size-fractionation chlorophyll-*a* measurements at the
shore-side, the frontal zone, and the seaside of the front to examine the spatial change of phytoplankton
physiology. Phytoplankton response to the river plume at the frontal zone was addressed by mixing the
local surface water with a varying percentage of plume water from the shore-side of the front. The
impact of river plume on the seaside of the front was further examined by incubations of the surface



seawater with the treatment of plume water. In addition to these experiments, the bottom water was
added to the surface water for incubation at various zones of the frontal system to estimate the impact of
vertical mixing or upwelling on surface phytoplankton community. We hope to use these experimental
approaches to address the responses of phytoplankton growth and physiology to the strong salinity front
over the shelf. Based on these field results, we will also discuss the impacts of river plume, vertical
mixing and coastal upwelling on physical and biogeochemical dynamics of the frontal systems in the
NSCS shelf-sea.

## 2. Material and methods

### 2.1 Description of the field work

Three field cruises aboard *R/V Zhanjiang Kediao* were performed during April, May, and June in 2016
with hydrographic and biogeochemical samplings over the NSCS shelf (Fig.1). A vertical transect
across the salinity front from the inner estuary to the shelf was intensively sampled during the June
(Section A in Fig. 1). There were three other transects (Section B, C, and D in Fig. 1) outside the PRE
with intense size-fractionation chlorophyll-*a* measurements during both May and June. Section B
transited across the frontal zone with Sections C and D on the seaside of the front. Surface waters at
different zones of the salinity front were selected for nutrient-enrichment experiments, including the
shore-side of the front (S1 and S2), the frontal zone (S3 and S4), and the seaside of the front (S5, S6 and
S7) during May and June 2016.

### 2.2 Measurements of hydrography, chlorophyll-a, nutrients and phytoplankton size structure

Seawater temperature, salinity, pressure, and fluorescence were acquired using a SeaBird model
SBE9/11 conductivity-temperature-depth (CTD) recorder and a Chelsea Aqua fluorometer. Discrete
water samples at 1m, 20m, 40m, 60m, 80m, and 100m were collected with Niskin bottles mounted onto
a Rosette sampling assembly (General Oceanic). After filtration onto a Whatman GF/F glass fiber filter,
the chlorophyll-*a* (Chl-*a*) sample was extracted by 90% acetone in darkness at 4 ℃ for 24 h and



determined using a Turner Design fluorometer (Knap et al., 1996). Three types of filters (20 μm Nylon
filter, 2 μm Polycarbonate filter, and 0.7 μm GF/F filter) were used to produce three different
size-classes including micro- (>20 μm), nano- (2-20 μm), and pico-phytoplankton (0.7-2 μm). Nutrient
samples were collected inline through a Whatman GF/F filter and frozen immediately at -20 ℃ until
analyzed. After thawing at room temperature, they were analyzed by an AA3 nutrient auto-analyzer
using colorimetric methods (Knap et al., 1996) with detection limits of 0.02, 0.02, and 0.03 μmol L$^{-1}$,
for nitrate plus nitrite (N+N), soluble reactive phosphate (SRP), and silicate (Si), respectively.

## 2.3 Setup of the ship-board incubation experiments

There were four different treatments prepared in duplicate for nutrient-enrichment experiments
including the control (C), nitrogen alone (+N), phosphorus alone (+P), and nitrogen plus phosphorus
(+NP). Nutrients were added to the incubation bottle to obtain final concentrations of 4.8 μM NaNO$_3$
and 0.3 μM NaH$_2$PO$_4$. Seawater samples were prescreened through a 200 μm mesh to remove large
grazers. These samples were incubated in 2.4 L transparent acid-cleaned polycarbonate bottles and
placed in a shipboard incubation chamber equipped with a flow-through seawater system. The incubator
was shaded to mimic 30% sunlight using a black filter with each bottle manually stirred twice a day.
Each incubation experiment lasted for two days with size-fractionated chlorophyll-*a* samples taken once
a day.
Surface water (~50L) collected at S2 outside the PRE mouth was saved as the plume water (PW).
Half of the PW was filtered through a 0.2 μm Millipore membrane filter (GTTP Isopore™) to produce
the filtered plume water (FPW). These waters were used to dilute the local surface waters at S6, S7 and
S8. Under the in-situ temperature and light, the mixture was incubated on board for two days with
size-fractionation chlorophyll-*a* collected each day. The bottom waters (BW) were collected at S2, S4,
S6 and S7 and stored in clean HDPE carboy. A 0.2-μm-filtration was used to create the filtered bottom
water (FBW). Both BW and FBW, with a final percentage of 12.5%, were added to the local surface
water for incubation to study the biological impact of vertical mixing or upwelling at these stations. We
also conducted a series of mixing experiments between surface waters of S2 and S4 with the final

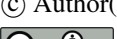

percentages of 0%, 25%, 50%, 75%, and 100% for S2, corresponding to the final salinity of 30.7, 24.7,
18.7, 12.7, and 6.6, respectively.
For each size class, the rate of daily chlorophyll-$a$ production ($\mu$g L$^{-1}$ d$^{-1}$) was calculated by the
difference of size-fractionated chlorophyll-$a$ concentration during each incubation day. We also
estimated the net growth rates $\mu$ (d$^{-1}$) for the water mixing experiment between S2 and S4 by $\mu$
$=ln(Chl_1/Chl_0)/\Delta t$, with $Chl_0$ and $Chl_1$ the initial and final size-fractionated chlorophyll-$a$ concentrations
every day ($\Delta t =1$ day). The specific growth rate approach could not work for other experiments, as large
errors of $\mu$ would arise when the initial chlorophyll-$a$ of a certain size-class ($Chl_0$) was close to zero.
**2.4 Estimations of horizontal advection and vertical mixing at the seaside of the front**
Assuming a salinity balance at the seaward front (Fong and Geyer, 2001), we have

$$U_e(S_0 - S) = K_H \frac{\partial S}{\partial z} \tag{1}$$

where $S$ and $S_0$ are salinity of the plume front and ambient water, $K_H$ is the eddy diffusivity, and the
bulk entrainment rate $U_e$ is computed by $U_e \approx 0.038 Ri^{-0.5}(\tau/\rho)^{0.5}$ with the Richardson number ($Ri$) given
by

$$Ri = \frac{g\rho}{\tau\rho_0} \int_0^h (\rho_0 - \rho)dz \tag{2}$$

with $g$ the gravitational acceleration, $\rho_0$ the ambient density, $h$ the thickness of plume front and $\tau$ the
wind stress (Fong and Geyer, 2001).
Horizontal nitrate flux to the surface water on the seaside of the front can thus be estimated by $J_h$
$=U_e(N-N_0)$ with $N$ and $N_0$ the nutrient concentrations of the plume front and the ambient water. The
vertical nitrate flux can be estimated by $J_v =K_H(\partial N/\partial z)$.



## 3 Results

### 3.1 Physical and biogeochemical setting of the NSCS shelf during the spring-summer

The temperature versus salinity diagram revealed a large change of hydrography during the three cruises (Fig. 2). There was a regional warming effect over shelf from April to June (Fig. 3A1-A3), along with the increase of wind strength (with a regional shift to upwelling favorable wind after the May, data no shown). The riverine input was clearly evidenced with low salinity waters in all the three cruises (Fig. 2). Spatially, there was a large area of low salinity in the coastal water west of the PRE (Fig. 3B1-B3), leading to a strong salinity front in the inner shelf. The plume water was mostly on the shore side of the front when the river-outflow flowing westward along the shore. The shore-side of the front was defined by a salinity of <26, the nearshore boundary of the plume (Wong et al., 2003), with the seaside of the front by a salinity of >32, the offshore boundary of the plume (Ou et al., 2007). The frontal zone is thus located in between the nearshore and offshore boundaries of river plume (Fig. 1).

In the coastal water west of the PRE, there was an intense chlorophyll-*a* bloom (Chl-*a* > 5 μg/L) on the shore-side of the front during all the three cruises (Fig. 3C1-C3), although the surface temperature of the bloom area increased from ~22$^{\circ}$C in April, to ~26$^{\circ}$C in May and to ~31$^{\circ}$C in June. The surface distributions of nitrate, silicate, and phosphate generally follow that of salinity for all the three cruises with much higher concentrations on the shore-side of the front than the seaside of the front (Fig. 3D and 3F). Interestingly, the surface low salinity tongue in the coastal water east of the PRE by eastward plume dispersion was cut off by a water mass of low temperature but high salinity during the June (Fig. 3A3 and 3B3). This water presumably should come from the subsurface via coastal upwelling, which was further supported by its higher phosphate concentration but lower N/P ratio compared to the ambient waters (Fig. 3D3 and 3F3).

There were substantial vertical changes of temperature, salinity, and chlorophyll-*a* while crossing the salinity front (Fig. 4A-4C) from the estuary to the shelf (Section A). The surface front was located in the inner shelf with the subsurface frontal zone going deep to the bottom of the estuary mouth (Fig.4A). Vertical distributions of nutrients generally followed that of salinity in the PRE with higher surface concentrations, whereas there was large drawdown of nutrients on the shore-side of the front





when approaching the edge of the river plume (Fig. 4D-4F), corresponding to a fast decrease of N/P
ratio from the shore-side of the front to the frontal zone. The dominant size-class shifted from micro- to
pico-cells while crossing the salinity front from the shore in Section B for both the May and June
cruises (Fig. 5). Variations in the percentages of micro- and nano-cells in Sections C and D were due to
a spatial change of the frontal zone (Fig. 5).

## 3.2 Variations of phytoplankton nutrient limitation over the NSCS shelf

Surface water properties of the incubation stations were summarized in Table 1. The highest
concentrations of nutrients and chlorophyll-*a* were in S1 and S2 on the shore-side of the front where
micro- and nano-cells dominated. A P-deficiency of the plume water can be inferred from the high N/P
ratios there. There was higher salinity (~30) but lower chlorophyll-*a* (~1 μg/L) in S3 and S4 at the
frontal zone, which should reflect a reduced impact of river plume. The surface waters of S5, S6 and S7
on the seaside of the front were dominated by pico-phytoplankton and showed the typical characteristics
of the open NSCS with low nutrients and chlorophyll-*a* but high salinity.
Phytoplankton total chlorophyll-*a* on the shore-side of the front (S1 and S2) and at the frontal zone
(S3 and S4) showed responses to P-addition but not N-addition, suggesting of P-limitation in these
waters (Fig. 6A-6D). In contrast, phytoplankton nutrient limitation varied substantially at S5, S6, and S7
on the seaside of the front (Fig. 6E-6G). Total chlorophyll showed no response to N-addition,
P-addition, and N-plus-P addition at S5 (Fig. 6E), which should suggest a relief of phytoplankton
community from N- and P-stresses there. There was a N-limitation of phytoplankton at S6, as the total
chlorophyll-*a* increasing with N-addition but not with P-addition (Fig. 6F), which was consistent with
its low N+N concentration of <0.5 μM at the surface (Table 1). Phytoplankton growth was P-limited at
S7 during the first day of incubation, but it became co-limited by both N and P during the second day of
incubation (Fig. 6G). This station (S7) was on the shelf edge, far away from the frontal zone, but was
influenced by the eastward extension of the plume as indicated by its relatively low surface salinity.
Interestingly, the response of phytoplankton total chlorophyll-*a* to nutrient treatments was mostly



mediated by micro-cells at stations S1, S2, and S3 where high nutrient concentrations and N/P ratios
were found (Fig. 6A2-6C2). In contrast, for stations S5, S6 and S7 on the seaside of the front, the
change of phytoplankton total chlorophyll-*a* at the surface layer was largely controlled by
pico-phytoplankton (Fig. 6D2-6G2). This result is consistent with the contention that larger
phytoplankton grow faster than small cell under nutrient replete conditions.

**3.3 Responses of surface phytoplankton to the addition of plume water**

The result of mixing experiments between the surface waters of S2 and S4 was shown in Fig.7. The
total chlorophyll-*a* was proportional to the amount of PW (the surface water of S2) in the mixture (Fig.
7A). As the PW has more chlorophyll-*a* than S4 (Table 1), the initial chlorophyll-*a* concentration of the
mixture showed a linear increase with the percentage of PW. The three phytoplankton size-classes
showed distinct responses to the ascending PW percentage during the first day of incubation (Fig. 7B).
There was a linear increase of the daily chlorophyll-*a* production rate of micro-cells with the percentage
of PW ($r^2$=0.9, $p$<0.01), whereas the production rate of nano-cells first increased with the PW
percentage from 0% to 50% and then remained relatively stable from 50% to 100%. Apart from both
micro- and nano-cells, pico-phytoplankton reached the maximal production rate at the 50% of PW
treatment. The responses of net growth rates to various PW treatments were slightly different from
those of the chlorophyll-*a* production rates (Fig. 7C). The net growth rate of micro-phytoplankton
increased with the PW percentage before becoming saturated at 75-100% PW. Pico-phytoplankton
showed a higher net growth rate but lower daily chlorophyll-*a* production rate than nano-phytoplankton
during the first day of incubation in the cases of 50-100% PW treatments. As the nutrients running out,
there were decreases of net growth rates for all the size-classes during the second day of incubation.
The chlorophyll-*a* biomass, as well as the daily chlorophyll-*a* production rate, of phytoplankton
was substantially enhanced by the addition of FPW at S6, S7, and S8 (Fig. 8). This should be expected
as the plume water had much more nutrients than the local waters on the seaside of the front. The
response of phytoplankton community to FPW was largely determined by nano- and pico-cells at these
stations. At station S6, the raw plume water (PW) was also added to the surface water for incubation to





account for the advective chlorophyll input by the river plume. Although the amount of PW added was
only 12.5%, it contributed about half of the chlorophyll biomass to the mixture for S6, which was due to
the high chlorophyll-*a* concentration of PW. That is why a stronger response of phytoplankton
chlorophyll-*a* to PW than to FPW was observed (Fig. 8A).
**3.4 Responses of surface phytoplankton to the addition of bottom waters**
The addition of FBW increased the total chlorophyll-*a* of S2, which was largely contributed by
micro-cells (Fig. 9A). At this station, the inclusion of FBW (lower N/P ratio) reduced the N/P ratio of
the surface water and thus the P-stress of surface phytoplankton. We found no difference in chlorophyll
responses to FBW and BW at S2, which could be due to the low chlorophyll-*a* of BW. Interestingly,
there was a net loss of phytoplankton chlorophyll-*a* with time at S4, which was not affected by the FBW
treatment (Fig. 9B). This is because nitrate and phosphate concentrations of the surface water were
similar to those of the FBW, although there was 9-fold increase of silicate in the FBW (Table 1). The
elevated silicate after FBW treatment did not stimulate a diatom growth given the sparse of micro-cells
in the surface water there. The addition of BW, however, substantially decreased the total chlorophyll-*a*
(Fig. 9B), likely reflecting the grazing activity in BW.
Phytoplankton growth was promoted by the FBW addition at S6 (Fig. 9C), as the N-stress of
phytoplankton could be relieved by the FBW with higher nitrate concentration and N/P ratio. We found
a reduced phytoplankton growth with the addition of BW compared to that of FBW (Fig. 9C), which
could also be attributed to the grazing activity of BW. At station S7, the BW was from the depth of 109
m with high nutrients but negligible chlorophyll-*a* compared to the surface water (Table 1). Therefore,
both FBW and BW additions increased surface phytoplankton growth (Fig. 9D). This stimulating effect
could be attributed to a reduced P-stress of phytoplankton in response to a lower N/P ratio of the surface
water.
**4 Discussion**





The persistent salinity front we observed from April to May of 2016 was a plume-induced buoyant front
(e.g. Ou et al., 2007), which could appear when the freshwater discharge was much stronger than the
tidal effect (Garvine and Monk, 1974). While governed by buoyancy, planetary rotation, and wind
forcing (Wong et al., 2003), the impact of the plume front on the coastal NSCS was large, as the low
salinity water spreading westward and eastward onto the large area of the shelf. A chlorophyll bloom on
the shore-side of the front was a direct response of phytoplankton to the river plume (Harrison et al.,
2008), as nutrient replenishment from the subsurface could be restricted by the salinity front with a
persistent stratification at the frontal zone. On the other hand, there was an intense upwelling found near
the coastal water east of the PRE, which could be due to an intensified cross-isobath transport of the
bottom boundary layer driven by an amplified alongshore current (Gan et al., 2009). Therefore, the
frontal system was affected by both river plume and coastal upwelling during the spring-summer of

12   2016.

13       Phytoplankton growth over the shore-side of the front was essentially P-limited, which is

consistent with previous findings (Zhang et al., 1999; Yin et al., 2001). Phytoplankton P-stress here is a
physiological response to the P-deficiency of the river plume due to the stoichiometric lack of P relative
to N (Moore et al., 2013). However, we found a spatial difference of phytoplankton physiology on the
seaside of the front, where there was less influence of river plume from the perspective of salinity.
Phytoplankton growth over the seaside of the front, dominated by small pico-cells, could be P-limited,
or N-limited, or not limited by N and P. There was no evidence of Si-limitation since micro-cell was not
stimulated by the filtered bottom water with a much higher silicate concentration. The spatial difference
of phytoplankton physiology is consistent with the nutrient variation of the developing plume front,
which should be regulated by both biological uptake and physical dynamics (Gan et al., 2014).

23       A balance between horizontal advection and vertical mixing can be approximately maintained at

the seaward front by an Ekman straining mechanism (Fong and Geyer, 2001) with salinity gradients
created by cross-shore Ekman current but destroyed by vertical mixing. Based on the hydrographic data,
we can estimate a horizontal entrainment rate $U_e$ of 0.5-1.0×10$^{-5}$ m/s and a vertical diffusivity $K_H$ of
0.8-1.7×10$^{-4}$ m$^2$/s across frontal boundary, which are comparable to those previously found over the





NSCS shelf (St. Laurent, 2008; Li et al., 2016). Horizontal nitrate flux to the seaside of the front is thus
0.2-3.6 mmolN/m$^2$/d. If we assume the same $K_H$ for the seaside of the front, we can also roughly
estimate a vertical nitrate diffusive flux of 0.6-4.7 mmolN/m$^2$/d, which is on the same order of
magnitude as the horizontal nutrient fluxes. Therefore, the varying nutrient supply driven by physical
dynamics, including cross-front advection and vertical mixing, might be responsible for the variability
of phytoplankton physiology on the seaside of the front.

7       The influence of river plume on the surface phytoplankton at the frontal zone was to directly

stimulate micro-phytoplankton growth, while a community P-limitation was still prevailing. Although
the growths of nano- and pico-cells were improved by low percentages of plume water (<50%), they
were inhibitated by high percentages of plume water (>50%). This finding is consistent with the
different nutrient uptake kinetics of the three phytoplankton size-classes (Finkel et al., 2009).
Micro-phytoplankton generally has a larger half-saturation constant for nutrient uptake than nano- and
pico-phytoplankton (Cermeno et al., 2005; Litchman et al., 2007). Therefore, small phytoplankton
(nano- and pico-cells) could become saturated with the ascending nutrients before micro-phytoplankton
did. At the frontal zone, nano-phytoplankton growth even well exceeded micro-phytoplankton at a low
percentage of plume water (<50%), which could explain the enhanced biomass percentage of nano-cells
at the frontal zone. On the seaside of the front, the plume-water addition (12.5%) indeed improved the
growths of all the three phytoplankton size-classes, regardless the type of nutrient limitation the surface
phytoplankton originally experienced.
Different from the shore-side of the front with a sharp decrease of nutrients at depths, the bottom
water on the seaside of the front showed much higher nutrient concentrations (but lower N/P ratios)
than the surface water, which was due to the intrusion of the deep water (Gan et al., 2014). Thus,
surface nutrient concentrations after vertical mixing or upwelling should decrease on the shore-side of
the front but increase on the seaside of the front. The final consequence of vertical mixing on both sides
of the front was to alter the N/P ratio of surface water closer to the Redfield ratio of 16 and thus
improved the growth of phytoplankton as showed in our experiments. While microplankton growth was
slightly stimulated by BW addition, our results on the seaside of the front did not show a shift of



phytoplankton community from pico- to micro-cells in response to upwelled nutrients from
deep-water-additions found in the western South China Sea (Cui et al., 2016) and in the open ocean
(McAndrew et al., 2007). In addition to nutrient stresses by varying nutrient concentrations and ratios,
phytoplankton growth at the frontal zone should also be influenced by other factors such as the change
of grazing pressure (Li et al., 2012). There were indeed evidences of enhanced grazing activity at
stations S4 and S6 when comparing incubation results of the filtered bottom water with those without
filtration. Therefore, a further study of grazing impact of zooplankton on various sizes of phytoplankton
and subsequent biomass accumulation at the frontal zone of the NSCS shelf may be a future research
priority. Since we have only focused on phytoplankton physiology of the surface layer, the future study
may also need to address the response of subsurface phytoplankton community to the frontal dynamics
over the shelf, since both the light field and nutrient conditions may vary substantially at the subsurface
layer across the salinity front.
**5 Conclusions**
Overall, the importance of physical-biological interaction in driving the patterns of phytoplankton
physiology and size-fractionated growths within a strong plume-induced salinity front over the NSCS
shelf was investigated by intense field measurements and shipboard incubation experiments during
April-June 2016. The current study suggested that variability of phytoplankton nutrient limitation and
size-fractionated growth on the shore-side, the seaside, and the frontal zone of the shelf-sea frontal
system could be attributed to varying nutrient supplies driven by physical dynamics of the frontal
system. While the impact of river plume was to directly increase the growth rates of all the three
phytoplankton size-classes, both nano- and pico-cells could become saturated with a high percentage of
plume water at the frontal zone. Vertical mixing or upwelling was found to substantially improve
surface phytoplankton growth over both sides of the front by altering the nutrient concentrations and
ratios. These results are important for a better understanding of physical control of coastal ecosystem
dynamics in the NSCS shelf-sea.



## Acknowledgements

We thank the captain and the staffs of *R/V Zhanjiang Kediao* for helps during the cruises. Drs Jie Xu and Dongxiao Wang were acknowledged for cruise assistants. This work is supported by the National Key Research and Development Program of China (2016YFC0301202) and the National Natural Science Foundation of China (41676108, 41706181).

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

| Station | Date | Depth [m]* | T [°C] | S [‰] | Chl-a [μg L⁻¹] | micro [%] | nano [%] | pico [%] | Si [μM] | N+N [μM] | SRP [μM] | N/P |
|---|---|---|---|---|---|---|---|---|---|---|---|---|
| S1 | 5/18 | 1 | 24.5 | 20.4 | 7.01 | 73.8 | 14.3 | 11.9 | 34.1 | 33.3 | 0.22 | 151 |
|    |      | 10 | 23.6 | 31.7 | 7.93 | 37.4 | 5.6 | 56.9 | 16.4 | 20.1 | 0.32 | 63 |
| S2 | 6/19 | 1 | 29.1 | 6.6 | 6.82 | 19.9 | 65.8 | 14.2 | 192.5 | 45.4 | 0.83 | 55 |
|    |      | 8 | 25.6 | 34.0 | 0.31 | 12.1 | 65.3 | 22.5 | 43.9 | 16.7 | 0.65 | 28 |
| S3 | 5/15 | 1 | 27.9 | 30.9 | 0.91 | 35.3 | 39.2 | 25.5 | 3.2 | 16.6 | 0.13 | 127 |
|    |      | 50 | 20.9 | 34.5 | 0.34 | 17.9 | 43.2 | 38.9 | 4.3 | 7.7 | 0.22 | 35 |
| S4 | 6/18 | 1 | 30.0 | 30.7 | 1.24 | 5.5 | 43.8 | 50.7 | 1.2 | 6.6 | 0.21 | 32 |
|    |      | 39 | 21.7 | 34.6 | 0.91 | 4.9 | 32.6 | 62.5 | 10.8 | 6.1 | 0.23 | 26 |
| S5 | 5/19 | 1 | 26.6 | 34.4 | 0.26 | 1.3 | 8.8 | 89.9 | 1.4 | 1.0 | 0.09 | 12 |
|    |      | 36 | 23.8 | 34.3 | 0.15 | 15.0 | 27.9 | 57.1 | 2.0 | 1.3 | 0.11 | 11 |
| S6 | 6/19 | 1 | 30.7 | 34.5 | 0.73 | 0.3 | 23.8 | 75.8 | 2.2 | 0.5 | 0.14 | 3 |
|    |      | 47 | 21.7 | 34.7 | 0.45 | 9.2 | 21.0 | 69.8 | 9.3 | 3.6 | 0.17 | 21 |
| S7 | 6/21 | 1 | 30.8 | 32.1 | 0.59 | 0.7 | 45.1 | 54.2 | 1.3 | 3.3 | 0.07 | 46 |
|    |      | 109 | 19.2 | 34.7 | 0.07 | 1.4 | 11.0 | 87.6 | 13.3 | 9.2 | 0.61 | 15 |

3    *The depth of surface water is always at ~1 m with the depth of bottom water 5-10 m above the topography.





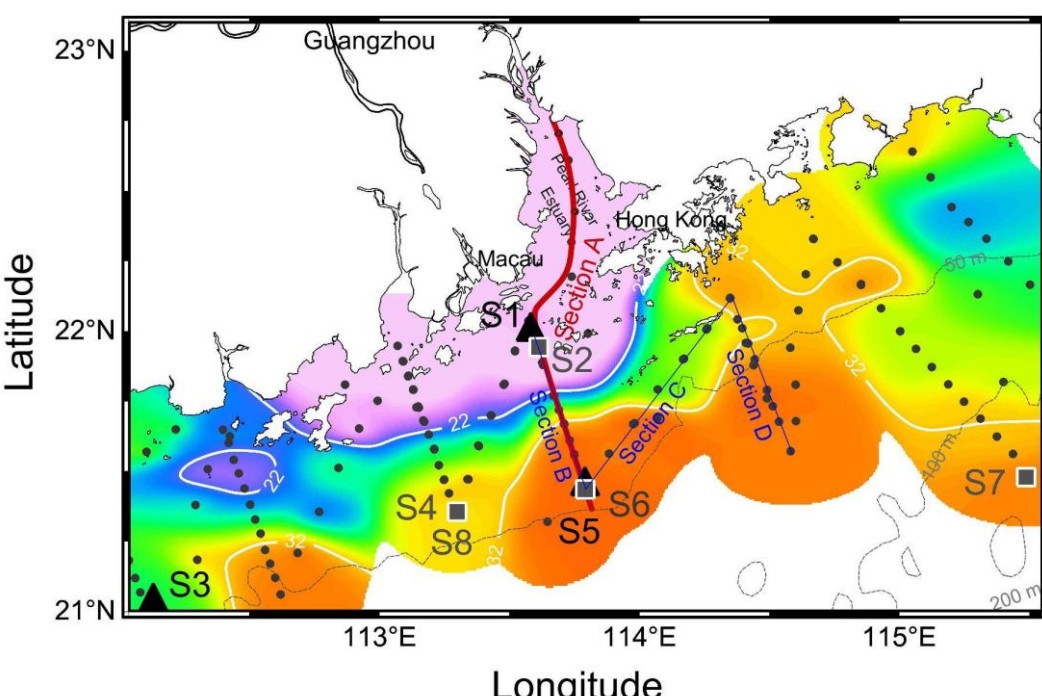

**Figure 1.** Sampling map in the NSCS shelf during May-June 2016. Color is the surface salinity with the

frontal zone by white lines of 26 and 32 (nearshore and offshore boundaries of the plume); Section A

across the front from the PRE to the shelf; section B across the front with sections C and D on the

seaside; triangles and squares are incubation sites S1-S7; dots are the stations with dash lines the

isobaths.



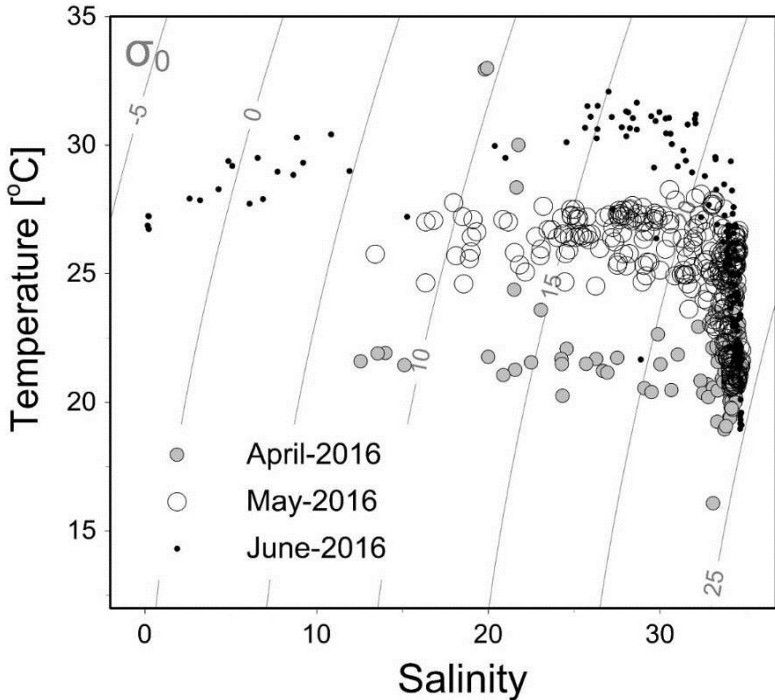

**Figure 2.** A Temperature vs. Salinity diagram during May-June 2016. Filled circles, open circles, and

dots are data of April, May and June cruises, respectively.



**Figure 3.** Surface distributions of **(A1-A3)** temperature, **(B1-B3)** salinity, **(C1-C3)** chlorophyll-*a*,
**(D1-D3)** nitrate, **(E1-E3)** silicate, and **(F1-F3)** phosphate in the NSCS during April, May, and June
2016. Small dots are the data points; open triangles and squares in B2-B3 show the positions of S1-S7.





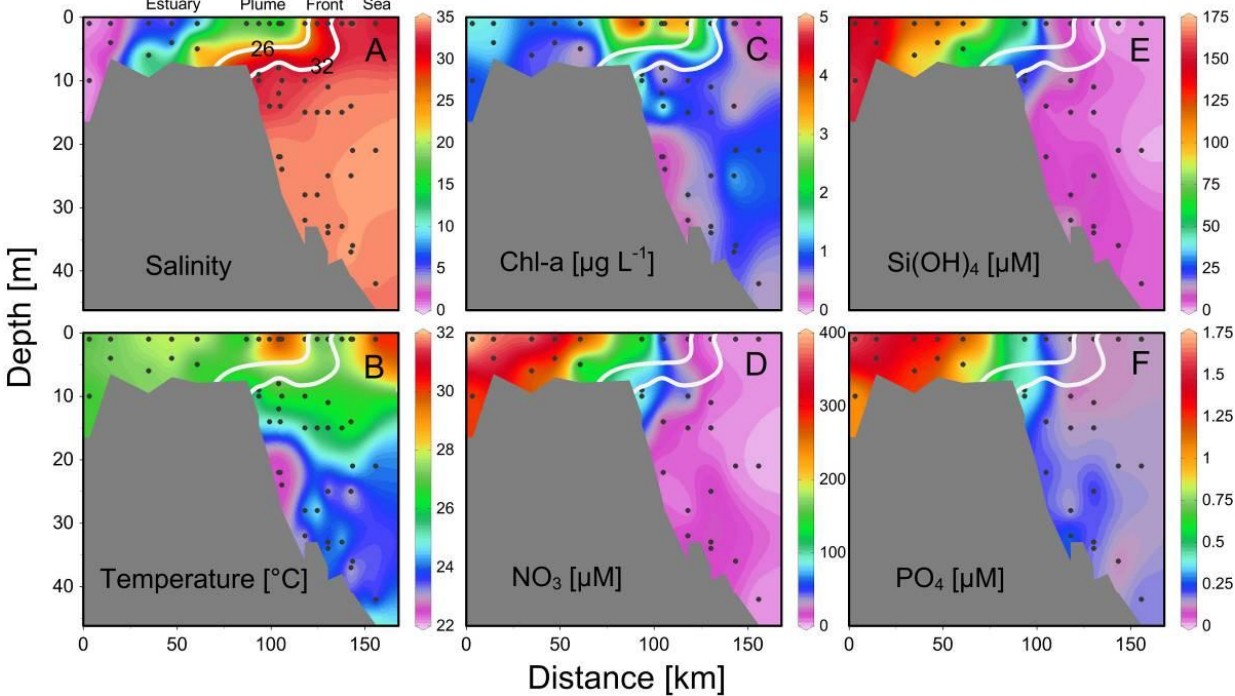

**Figure 4.** Vertical distribution of **(A)** salinity, **(B)** temperature, **(C)** chlorophyll-*a*, and **(D)** nitrate, **(E)**
silicate, and **(F)** phosphate across the front from the estuary to the sea. Location of the section is in
Fig.1. Two white lines overlaid are salinity of 26 and 32 for nearshore and offshore boundaries of the
plume (see text for detail).





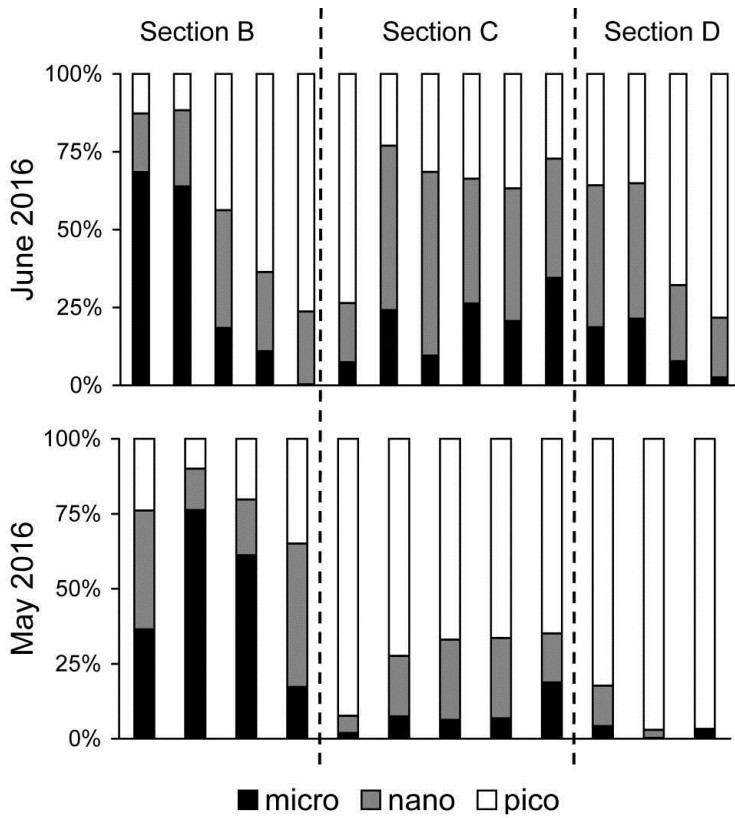

**Figure 5.** Comparisons of size-fractionation chlorophyll-*a* for sections B, C, and D between May and June 2016.





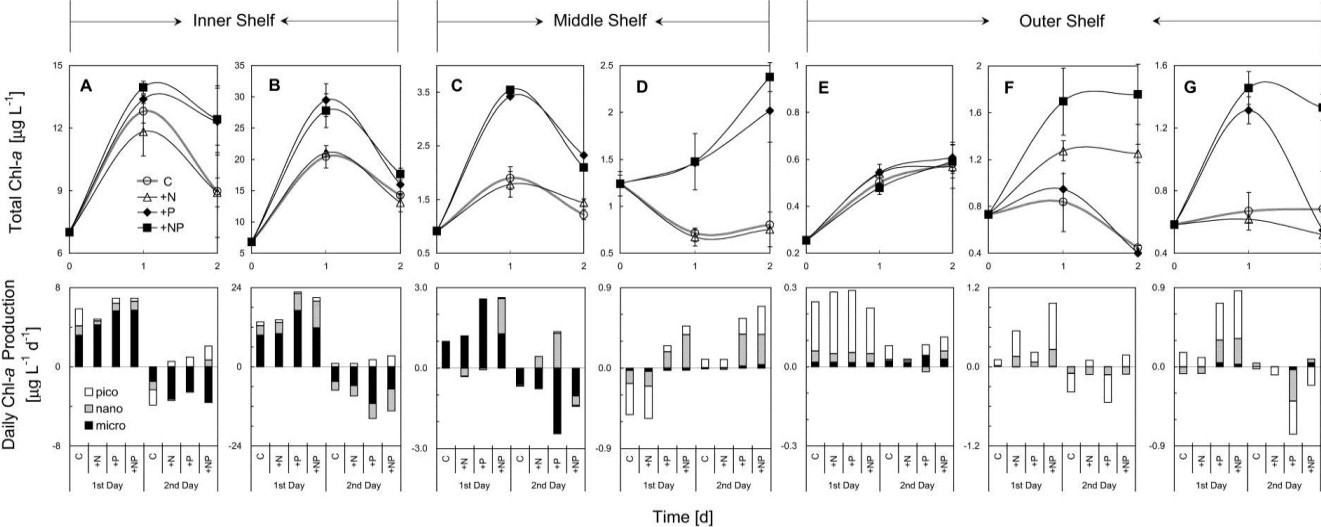

**Figure 6.** Responses of total chlorophyll-*a* and size-fractionated daily chlorophyll-*a* production rate of the surface water to various nutrient enrichments at **(A)** S1, **(B)** S2, **(C)** S3, **(D)** S4, **(E)** S5, **(F)** S6, and **(G)** S7 during May and June 2016. Station locations are in Figure 1 with the initial conditions in Table 1; Treatments include control (C), nitrogen alone (+N), phosphorus alone (+P), and nitrogen plus phosphorus (+NP), respectively.





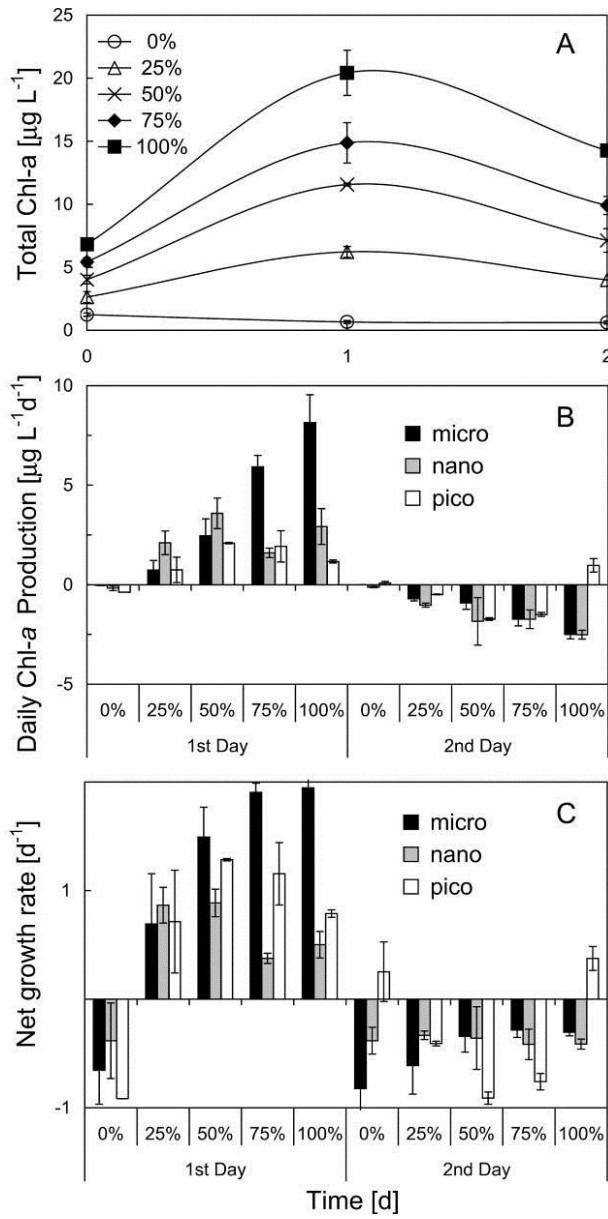

3  **Figure 7.** Responses of **(A)** total chlorophyll-*a*, **(B)** size-fractionated rate of daily chlorophyll-*a*

4  production, and **(C)** size-fractionated net growth rate of the surface water at S4 to a various percentage

5  of plume water from S2.



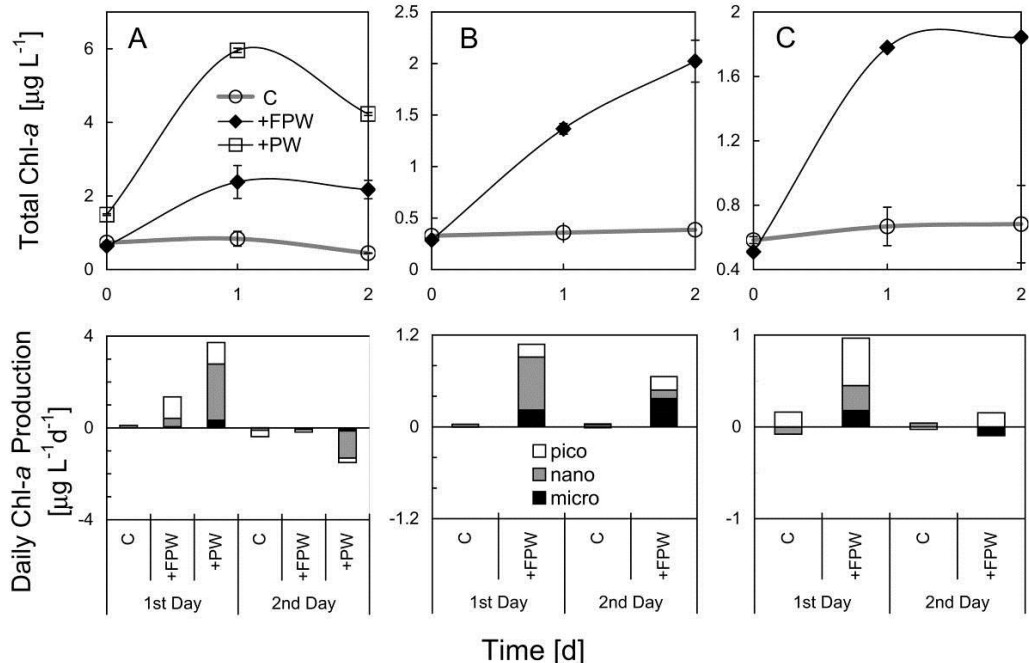

**Figure 8.** Responses of total chlorophyll-*a* and size-fractionated rate of daily chlorophyll-*a* production of the surface water to the addition of plume water at **(A)** S6, **(B)** S7, and **(C)** S8. PW is the plume water with FPW the filtered plume water.

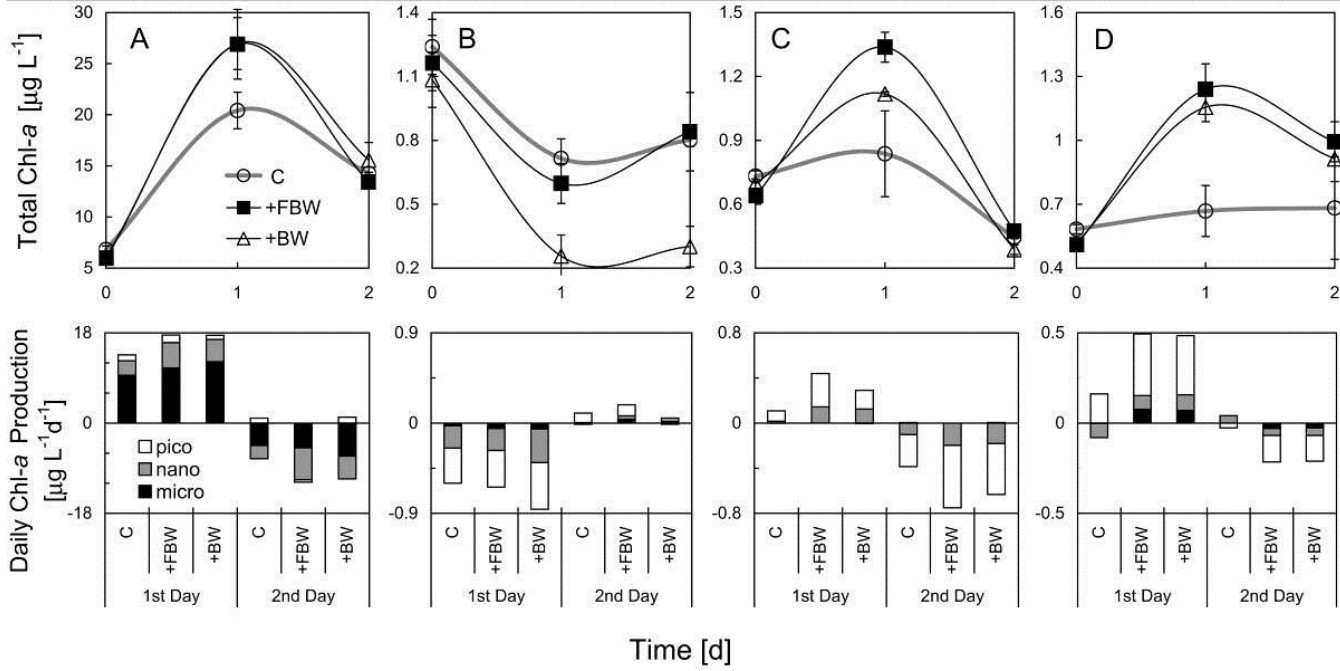

**Figure 9.** Responses of total chlorophyll-*a* and size-fractionated rate of daily chlorophyll-*a* production

of the surface water to the addition of local bottom waters at station **(A)** S2, **(B)** S4, **(C)** S6, and **(D)** S7

during June 2016. BW is the bottom water with FBW the filtered bottom water.