# Peer review of "Phytoplankton response to a plume front in the northern South China Sea"

_Biogeosciences, 2017_

## Short Comment (SC1) · 9 Jan 2018

The authors have studied the impact on phytoplankton population growth and community size structure from the shore side to the sea side of a strong salinity front, set up by freshwater discharge from the Pearl River Estuary (PRE), over the shelf waters in the northern South China Sea. Apart from the extensive hydrographical study over the area impacted by the river plume, they also conducted several experiments at different stations along the PRE to marine gradient. These were designed to assess phytoplankton nutrient limitations, by the addition of inorganic nitrogen and phosphorus, as well as the influence on productivity and community structure of either vertical (upwelling) or horizontal mixing (river discharge) of water masses.

[Figure]

They found that the region studied was highly dynamic over the months studied, but that the phytoplankton community showed P-limitation at, and shore-side, of the salinity front, whereas sea-side waters tended to be N-limited. However, upwelling of sea-side bottom water would promote growth in the surface populations on both sides of the front as suggested by the mixing experiments. Discharge of plume water also promoted growth of the larger cells. The authors conclude that the physical dynamics of the river plume could deeply affect phytoplankton physiology and growth.

This is predominantly a well written manuscript with mostly smaller points and edits. However, I do have two main criticisms 1) that nutrient concentrations were not measured from the incubation experiments to assess nutrient drawdown over time, and 2) the design of the plume water mixing experiment. Having the nutrient's fate in the incubations may have shed light on if the loss of chlorophyll at day 2 was due to nutrient limitation, or something else. This would have added much value to this study. The design of the plume water mixing experiment makes it difficult to interpret the relative importance of the seed community (structure, biomass) and the influence of nutrients and salinity changes. As it is presented here I do not think the results support the conclusions drawn.

The mixing of whole plume water with whole 'sea-side' water at different ratios, is in effect a dilution of one community by the other, with the 100% and 0% being the end-members. From data in Fig 7A, the chlorophyll based growth rates ($\mu$) are more of less identical for all additions of plume water, indicating that the results are reflecting the dilution, not changes in growth rates, even if the final chl a is increasingly higher with higher plume water addition. I suggest that the authors carefully reexamine this experiment and its outcome in a revision of this manuscript.

Other comments:

P3, ln14. Suggest adding Mahaffey et al. (2012) here. This paper contains data on mixing experiments too that may be informative for this manuscript too. (Mahaffey et

al., 2012. Phytoplankton response to deep seawater nutrient additions in the North Pacific Subtropical Gyre. MEPS 460:13-34)

P4, ln 18,19. What about station 8? Should it be listed here too? (p5, ln 22 ?)

P5, ln 1-3. Choice of filter types. Why were the filters of such different materials? Do they have different retention characteristics apart from pore size? Where the chlorophyll fractions determined by difference or where these from sequential filtrations?

P5, ln 4. Did you see any Si contamination from the glass fiber filter used?

P5, ln 13. What determined the final concentrations of N and P added? Perhaps also add that these are at ~ 16:1 or Redfield ratio for N:P.

P5, ln18. Suggest adding at what stations and at what dates these N+P nutrient addition experiments were performed.

P6, ln 1. Please describe what question this experimental design was meant to answer, or test? Also, please add when these where performed.

P7, ln 18-20. This sentence is confusing to me. What is meant by "..east of the PRE by eastward plume dispersion.."? That the low salinity tongue from the PRE was cut off by another water mass with low temperature and high salinity?

P9 – plume water mixing experiment. This is my main problem with this manuscript. The design of this experiment does not allow for testing what I think was the intention to test. Which is to say, the effect on plume water mixing (with its extant community and nutrients) with seaside water (with its extant community and nutrients). However, the way this is set up, it is difficult to separate what changes in chl is derived from the seed population or the changes in available nutrients. This would have needed to also include reciprocal dilutions using filtered PW and/or surface seawater.

P9, ln 21. Are the nutrients running out? Are there data to show this?

P12, ln 10. Have the effect of changing salinity on phytoplankton growth for the sea

side versus plume water plankton been considered?

P13, ln 3. Suggest citing Mahaffey et al 2012 here too

P18 Table 1. Should data from station 8 be included here too?

Figures 6-9. It would be helpful to see the chl concentration of the size fractions at t0 in these graphs. Also, it would be good to add when each of these experiments were carried out.

End review

---

## Referee Comment (RC2) · Anonymous Referee #2 · 25 Jan 2018

The story about the enhanced micro-phytoplankton growth by elevated nutrients is not new. But it's lucky, in the northern South China Sea, we have large river discharge carried with high nutrient concentration (33.3 $\mu$M N+N in S1) and extremely high N/P ratio (151:1), the upwelling (with N/P close to the Redfield ratio) also contributes to the nutrient dynamics in the northern SCS shelf. It's a perfect place to examine both the effects of increasing nutrient concentrations and altering N/P ratios. I like the statement "the mixing of bottom water would stimulate the growth of surface phytoplankton on both sides of the front by altering the surface N/P ratio closer to the Redfield stoichiometry". It's like a shelf-tuning mechanism of marine ecosystem to cope with the anthropogenic influences by increasing N inputs. The authors conducted the comprehensive and heavy-work experiments to examine the phytoplankton dynamics during

the mixing processes between non-plume waters and plume waters and bottom waters (representing upwelling). For the front, the mixing is of both nutrients and phytoplankton; but for the sea-side, the mixing is only of nutrients; it's reasonable because of the distances. And the manuscript is well written.

The discussion is insufficient to address all the issues. It's not surprise to see enhanced phytoplankton growth by elevated nutrients. But what about the N/P ratio? The increasing Chla production and net growth rate with mixed ratio in Fig. 7 in day 1 is not the response of S4 phytoplankton to PW addition, but essentially the response of mixed communities of S2 and S4 to different levels of nutrients concentrations and N/P ratios (Fig. 1). The differences in net growth rate may be related to the changes of N/P ratio. The discussion about the optimal N/P ratios for phytoplankton in plume waters is essential (Geider & Roche 2002). If we compare Fig. 8 with Fig. 9 and Fig. 6 for the same stations, such as S6 and S7, we can find different changes in size structure of Chla in day 1. Nano-phytoplankton showed greater responses to FPW than to BW/FBW and to nutrient addition. It may imply influences of N/P ratio on different species.

I find the incubations were made in May and June. But the manuscript has the April hydrographical data. It confuses me. What's the additional value of April data to this paper? And there are some questions about your graphs:

Fig. 1. You may zoom out a little bit, so we can see S3 and S7 clearly. You have the salinity contour in the graph. Is it a composite of three cruises or just the June cruise? You use different symbols for S1, S3, S5 and S2, S4, S6, S7, respectively. Do you mean S1, S3, S5 were in May, S2, S4, S6, S7 were in June? Were S5 and S6 in the same position? What is S8 in the map? You don't have S8 in Table 1.

Fig. 2. The title is "A Temperature vs. Salinity diagram during May-June 2016". But you have April data in it.

Fig. 4. The same question, is it a composite of three cruises or just the June cruise?

Other comments:

P2-21, "affect the large area of" is "affect a large area"

P2-25, "a P-limitation of phytoplankton" is "a P-limitation of phytoplankton growth"

P5-16, What's "black filter", do you mean "neutral filter"?

P5-21, "These waters were used to dilute the local surface waters at S6, S7 and S8" in what percentage?

P5-26, can you specify the "biological impact"?

P6-1, The percentage of 100% for S2, literally means no S4 waters, but you said it's a mixing experiment. It's good to have a comparison. But it's confusing. Maybe to say S2 instead of 100%.

P7-4, "warming effect" is usually used when we are referring to an impact of global change, here I think it's just a seasonal change of temperature.

P7-20, "This water", I can see two water masses in the previous sentence. So, which one did you refer to?

P9-3, "controlled by" is "contributed by"

P9-14-17, I can see a smaller value of nano-phytoplankton chla in 75% than that in 50%. Does pico-phytoplankton chla in 75% statistically lower than that in 50%?

P9-27 & P10-1, "At station S6, the raw plume water (PW) was also added to the surface water for incubation to account for the advective chlorophyll input 1 by the river plume." This necessary information should be in the method section.

P10-8, you can do the maths for N/P ratios since you have the numbers in the table.

P10-6, in the section 3.4, the discussion about the nutrient limitation status and grazing activity should be moved to the discussion section.

[Figure]

| | Shore-side | Front | Sea-side | Closer to Redfield N/P ratio? |
|---|---|---|---|---|
| +4.8 μM NaNO3 +0.3 μM NaH2PO4 | S1, S2 | S3, S4 | S5, S6, S7 | Yes |
| +PW | | S4, mixed with nutrients and phytoplankton of S2 | S6, mixed with nutrients and phytoplankton of S2 | Front: yes for S2 phytoplankton, no for S4 phytoplankton; Sea-side: yes for S2 phytoplankton, yes or no for S4 phytoplankton depending on mixed ratio |
| +FPW | | | S6, S7, S8, mixed with nutrients of S2 | Yes for S6 |
| +BW | S2, grazing effects? | S4, grazing effects? | S6, S7, grazing effects? | Yes |
| +FBW | S2, no grazing effects? | S4, no grazing effects? | S6, S7, no grazing effects? | Yes |

**Fig. 1.** Summary of incubation experiments

---

## Referee Comment (RC3) · Anonymous Referee #3 · 26 Jan 2018

The manuscript investigated the responses of phytoplankton growth and community size structure to a plume front through hydrographic measurements and nutrient-enrichment experiments in the coastal water west of Pearl River Estuary over the Northern South China Sea shelf. Field surveys from spring to summer indicated that the frontal system was affected by both river plume and coastal upwelling through analyzing temperature, salinity, etc. Some field experiments were designed to assess nutrient limitation for phytoplankton growth by addition of inorganic nutrients, and the influences of plume water and upwelling on phytoplankton community structure and productivity. They found that phytoplankton productivity on the shore-side of the front showed P-limitation, while N-limitation on the seaside. Plume waters and bottom waters would largely contribute to the phytoplankton productivity and impact

community size structure. The authors have done many works to implement the study of phytoplankton growth responses to plume front, while they just simply summarized what they did without comprehensive discussion. I do have two main criticisms: 1) the nutrient concentrations were not measured to assess nutrients uptake by phytoplankton in the shipboard incubation experiments, and 2) the incubation bottles with smaller volume. The phytoplankton in culture media with smaller volume would be diluted by addition of plume waters and bottom waters, and the water sample could not be enough to get chl a samples. I do not think incubation experiments lasted for two days was enough to evaluate the phytoplankton growth to inorganic nutrients because the culture time is too short. Other comments: 1) In the manuscript, there were no parameters concerning physiological response. 2) P6, line 1-2, the descending salinity would have obvious effect on phytoplankton growth, and the paper didn't evaluated the direct effects of salinity. 3) P8 line 16 delete "of" 4) The incubation site S8 was not marked in Figure 1. The hydrographic and biogeochemical properties of S8 were not mentioned too. 5) In Figure 1 the white salinity lines were marked as 22 and 32, which were described as 26 and 32. 6) In Figure 2 "A Temperature vs. Salinity diagram during May-June 2016" should be corrected as April-June 2016.

Please also note the supplement to this comment:
https://www.biogeosciences-discuss.net/bg-2017-495/bg-2017-495-RC3-supplement.pdf

———————————————

---

## Author Comment (AC1) · 9 Mar 2018

Response to Reviewer #1 (by K. Bjorkmanïïjĺ 1. "... However, I do have two main criticisms 1) that nutrient concentrations were not measured from the incubation experiments to assess nutrient drawdown over time. Having the nutrient's fate in the incubations may have shed light on if the loss of chlorophyll at day 2 was due to nutrient limitation, or something else. This would have added much value to this study."

Response: Nutrient concentrations were actually measured during incubation experiments. We have added these data to the revised manuscript to discuss the changes of chlorophyll and nutrients over incubation time. P-limitation at stations S1, S2, S3, and

[Figure]

S4 was confirmed by change of N and P during incubations: there were enhanced N consumptions by addition of P, but P consumptions were not stimulated by addition of N. For S6, N-limitation was supported by enhanced P consumptions by N-addition (but N consumption was not enhanced by P-addition). The nutrient data also suggested that N and P co-limitation at the second day of incubation found at S7 was due to running out of N by P-addition during the first day of incubation.

2. "2) The design of the plume water mixing experiment. The design of the plume water mixing experiment makes it difficult to interpret the relative importance of the seed community (structure, biomass) and the influence of nutrients and salinity changes. As it is presented here I do not think the results support the conclusions drawn. The mixing of whole plume water with whole 'sea-side' water at different ratios, is in effect a dilution of one community by the other, with the 100% and 0% being the end members. From data in Fig 7A, the chlorophyll based growth rates ($\mu$) are more of less identical for all additions of plume water, indicating that the results are reflecting the dilution, not changes in growth rates, even if the final chl a is increasingly higher with higher plume water addition. I suggest that the authors carefully reexamine this experiment and its outcome in a revision of this manuscript."

Response: We agree with the reviewer that the design of the plume water mixing experiment cannot separate the different effects by the changes of nutrients, salinity, and phytoplankton population. However, we don't intend to separate these factors by the plume water mixing experiments. What we focus on is the combined effects of varying community, salinity and nutrients during the mixing experiments, given the relatively short distance of these two waters at the frontal zone. To separate the effects of nutrients and seed population inputted by PW, we had to chose a small percentage of FPW addition (12.5%) at station S6 to ensure that the initial chl-a concentration after FPW treatment is comparable to that of the control experiment (Figure 8A1-A4). In that case, the difference between the FPW treatment and the control should reflect the impact of nutrients, whereas the difference between PW and FPW treatments should

reflect the impact of seed population inputted by PW. The chlorophyll-based community net growth rate ($A$=ln(Chl_t/Chl_0)) is -0.6 d-1 for S4 (local water at the front), but it is 1.0 d-1 for S2 (the plume water). Since the initial chl-a concentration of S2 was about 6 time of that of S4, the mixed community (for 25%, 50%, 75% PW treatments) should be dominated by population from the plume community. That is why we can see a similar positive net growth rate for all additions of plume water. On the other hand, the apparent net growth rate ($A$) should be determined by the specific growth rate (A) and the grazing mortality rate (B) based on the equation of $A$=A-B (e.g. Landry and Hassett 1982). Therefore, if the growth and grazing of phytoplankton are tightly coupled during the dilution, we should not expect to see a large change of net growth rate. We have carefully discussed these results in the revised manuscript.

3. P3, ln14. Suggest adding Mahaffey et al. (2012) here. This paper contains data on mixing experiments too that may be informative for this manuscript too. (Mahaffey et al., 2012. Phytoplankton response to deep seawater nutrient additions in the North Pacific Subtropical Gyre. MEPS 460:13-34)

Response: Done.

4. P4, ln 18, 19. What about station 8? Should it be listed here too? (p5, ln 22?)

Response: S8 is located at the same place as S4 but at different time, which had already been clearly stated in Table 1.

5. P5, ln 1-3. Choice of filter types. Why were the filters of such different materials? Do they have different retention characteristics apart from pore size? Where the chlorophyll fractions determined by difference or where these from sequential filtrations?

Response: The three types of filters have all been previously used for phytoplankton size-fractionation in the South China Sea (Huang et al., 2008; Chen et al., 2009). Chlorophyll fractions were determined by sequential filtrations using these filters during our cruises. The GF/F filter (0.7 $\mu$m) can be used for collecting picoplankton due to its

high particle absorption ability leading to similar retention ability as 0.2 $\mu$m Millipore membrane filter.

Huang et al., Spatial and temporal distribution of nanoflagellates in the northern South China Sea, Hydrobiologia, 605:143–157, 2008. Chen et al., Trophic interactions within the microbial food web in the South China Sea revealed by size-fractionation method, Journal of Experimental Marine Biology and Ecology, 368: 59–66, 2009;

6. P5, ln 4. Did you see any Si contamination from the glass fiber filter used?

Response: We did not see Si contamination for GF/F filter. Silicate concentration after GF/F filtration was not much different from that after 0.2 $\mu$m membrane filter.

7. P5, ln 13. What determined the final concentrations of N and P added? Perhaps also add that these are at ~16:1 or Redfield ratio for N:P.

Response: Thanks for pointing out this. The final In the revised manuscript, we have clearly state that the additions of N and P for incubation experiments were based on the Redfield N:P ratio of 16 to 1.

8. P5, ln18. Suggest adding at what stations and at what dates these N+P nutrient addition experiments were performed.

Response: Done.

9. P6, ln 1. Please describe what question this experimental design was meant to answer, or test? Also, please add when these where performed.

Response: Done. The mixing experiment conducted on June 19th, 2016 is to simulate phytoplankton response to the intense mixing process by the dispersive river plume.

10. P7, ln 18-20. This sentence is confusing to me. What is meant by "...east of the PRE by eastward plume dispersion..."? That the low salinity tongue from the PRE was cut off by another water mass with low temperature and high salinity?

Interactive
comment

Response: We are sorry for confusing. We have rewritten these sentences in the revised manuscript. The surface low salinity tongue in the coastal water east of the PRE (generated by eastward plume dispersion) was cut off by another water mass of low temperature but high salinity during the June

11. P9 – plume water mixing experiment. This is my main problem with this manuscript. The design of this experiment does not allow for testing what I think was the intention to test. Which is to say, the effect on plume water mixing (with its extant community and nutrients) with seaside water (with its extant community and nutrients). However, the way this is set up, it is difficult to separate what changes in chl is derived from the seed population or the changes in available nutrients. This would have needed to also include reciprocal dilutions using filtered PW and/or surface seawater.

Response: The reviewer is right about that our design of the mixing experiment between S2 and S4 could not separate the effects between seed population and nutrients. Actually, we don't intend to separate these two as they should be both important for phytoplankton chl-a change at the frontal zone given the relatively short distance of the two waters. The including of reciprocal dilution experiments with the filtered plume water (FPW) and/or filtered surface sweater of S4 cannot separate the effect of varying nutrients from that of the change of seed phytoplankton. The reason is that the initial chl-a concentration will be largely diluted along with the increase of nutrients. To separate the effects of nutrients and seed population inputted by PW, we had chosen a small percentage of FPW addition (12.5%) at station S6 to ensure that the initial chl-a concentration after FPW treatment is comparable to that of the control experiment (Figure 8A1-A4). In this case, the difference between the FPW treatment and the control should reflect the impact of nutrients, whereas the difference between PW and FPW treatments should reflect the impact of seed population inputted by PW.

12. P9, ln 21. Are the nutrients running out? Are there data to show this?

Response: Yes, phosphate was almost running out during the second day of incubations. We have added nutrient data to the revised manuscript.

13. P12, ln 10. Have the effect of changing salinity on phytoplankton growth for the sea side versus plume water plankton been considered.

Response: We have added discussions of the impact of salinity on phytoplankton growth in the revised manuscript. Coastal phytoplankton species can generally tolerate a much larger range of salinity than estuarine and oceanic species (e.g. Brand 1984). The salinity of 6.6-30.7 during the mixing experiment at the frontal zone is higher than the lethal level of ~5 for most estuarine phytoplankton species due to osmotic pressure (Kies, 1997; Floder et al., 2010). However, inter-specific differences in salinity tolerances of phytoplankton may be important for phytoplankton growth at the lower ranch of the PRE where fluctuating salinities between 0-10 were found.

14. P13, ln 3. Suggest citing Mahaffey et al 2012 here too

Response: Done.

15. P18 Table 1. Should data from station 8 be included here too?

Response: Done. Data of station S8 was added to table 1.

16. Figures 6-9. It would be helpful to see the chl concentration of the size fractions at t0 in these graphs. Also, it would be good to add when each of these experiments were carried out.

Response: We decided to not show the initial size-fractionated chl-a data in these Figures. The initial size-fractionated chl-a concentrations for S1-S8 (Figure 6) had already been shown in Table 1. For other experiments in Figures 7-9, the initial size-fractionated chl-a concentrations were simply calculated based the fractions of waters mixed for these stations. The start dates of incubation experiments have been added to figures in the revised manuscript.

Please also note the supplement to this comment:

https://www.biogeosciences-discuss.net/bg-2017-495/bg-2017-495-AC1-supplement.pdf

[Figure]

**Supplement:**

[revised manuscript text omitted]

---

## Author Comment (AC2) · 9 Mar 2018

Response to Reviewer #2 (25 January 2018)

1. "The discussion is insufficient to address all the issues. It's not surprise to see enhanced phytoplankton growth by elevated nutrients. But what about the N/P ratio? The increasing Chla production and net growth rate with mixed ratio in Fig. 7 in day 1 is not the response of S4 phytoplankton to PW addition, but essentially the response of mixed communities of S2 and S4 to different levels of nutrients concentrations and N/P ratios (Fig. 1). The differences in net growth rate may be related to the changes of N/P ratio. The discussion about the optimal N/P ratios for phytoplankton in plume waters is essential (Geider & Roche 2002). If we compare Fig. 8 with Fig. 9 and

[Figure]

Fig. 6 for the same stations, such as S6 and S7, we can find different changes in size structure of Chla in day 1. Nano-phytoplankton showed greater responses to FPW than to BW/FBW and to nutrient addition. It may imply influences of N/P ratio on different species."

Response: In the revised manuscript, we have added more discussions related to the influence of N/P ratio on phytoplankton chl-a production and growth rate. We agree with the reviewer that the mixing experiment shown in Figure 7 should reflect the response of the mixed community (S2 and S4) to varying mixing conditions (with different nutrient concentrations and N/P ratios). We have discussed the impact of the optimal N/P ratios on the different phytoplankton species as suggested by the reviewer.

2. "I find the incubations were made in May and June. But the manuscript has the April hydrographical data. It confuses me. What's the additional value of April data to this paper?"

Response: We think the hydrographic data of April cruise is valuable to this manuscript, since it can better present the temporal change of the plume front during the spring-summer.

3. "Fig. 1. You may zoom out a little bit, so we can see S3 and S7 clearly. You have the salinity contour in the graph. Is it a composite of three cruises or just the June cruise? You use different symbols for S1, S3, S5 and S2, S4, S6, S7, respectively. Do you mean S1, S3, S5 were in May, S2, S4, S6, S7 were in June? Were S5 and S6 in the same position? What is S8 in the map? You don't have S8 in Table 1."

Response: Done. We have redone Figure 1 as suggested by the reviewer. It is a composite of three cruises. S1, S3, and S5 are from May and S2, S4, S6, S7, and S8 are from June. S8 and S6 are located in the same places as S4 and S5, respectively. We have also added S8 to Table 1.

4. "Fig. 2. The title is "A Temperature vs. Salinity diagram during May-June 2016". But

you have April data in it."

Response: Done. Thanks for pointing out this. We have corrected it the revised manuscript.

5. Fig. 4. The same question, is it a composite of three cruises or just the June cruise?

Response: It is a composite of three cruises.

6. P2-21, "affect the large area of" is "affect a large area"

Response: Done.

7. P2-25, "a P-limitation of phytoplankton" is "a P-limitation of phytoplankton growth"

Response: Done.

8. P5-16, what's "black filter", do you mean "neutral filter"?

Response: Yes, it is a neutral filter. We have corrected this in the revised manuscript.

9. P5-21, "These waters were used to dilute the local surface waters at S6, S7 and S8" in what percentage?

Response: The percentage of dilution was 12.5%. We have added this information to the paragraph in the revised manuscript.

10. P5-26, can you specify the "biological impact"?

Response: Done. It is the impact of vertical mixing and upwelling on phytoplankton growth. We have clarified this in the revised manuscript.

11. P6-1, the percentage of 100% for S2, literally means no S4 waters, but you said it's a mixing experiment. It's good to have a comparison. But it's confusing. Maybe to say S2 instead of 100%.

Response: Done.

12. P7-4, "warming effect" is usually used when we are referring to an impact of global change, here I think it's just a seasonal change of temperature.

Response: Done. We have replaced it by "increase of temperature"

13. P7-20, "This water", I can see two water masses in the previous sentence. So, which one did you refer to?

Response: It is the water of low temperature and high salinity. We have clarified these in the revised manuscript.

14. P9-3, "controlled by" is "contributed by"

Response: Done.

15. P9-14-17, I can see a smaller value of nano-phytoplankton chla in 75% than that in 50%. Does pico-phytoplankton chla in 75% statistically lower than that in 50%?

Response: We do not found statistical difference for picophytoplankton between 50% and 75%. We have corrected this in the revised manuscript.

16. P9-27 & P10-1, "At station S6, the raw plume water (PW) was also added to the surface water for incubation to account for the advective chlorophyll input by the river plume." This necessary information should be in the method section.

Response: Done.

17. P10-8, you can do the maths for N/P ratios since you have the numbers in the table.

Response: Done.

18. P10-6, in the section 3.4, the discussion about the nutrient limitation status and grazing activity should be moved to the discussion section.

Response: Done.

---

## Author Comment (AC3) · 9 Mar 2018

Response to Reviewer # 3 (26 January 2018)

1. The nutrient concentrations were not measured to assess nutrients uptake by phytoplankton in the shipboard incubation experiments

Response: Nutrient concentrations were actually measured during incubation experiments. We have added these data to the revised manuscript to discuss the changes of chlorophyll and nutrients over incubation time. P-limitation at stations S1, S2, S3, and S4 was confirmed by change of N and P during incubations: there were enhanced N consumptions by addition of P, but P consumptions were not stimulated by addition of N. N-limitation of S6 was supported by enhanced P consumptions by N-addition (but

N consumption was not enhanced by P-addition). N and P co-limitation at the second day of incubation found at S7 was due to running out of N by P-addition during the first day of incubation.

2. The incubation bottles with smaller volume. The phytoplankton in culture media with smaller volume would be diluted by addition of plume waters and bottom waters, and the water sample could not be enough to get chl a samples. I do not think incubation experiments lasted for two days was enough to evaluate the phytoplankton growth to inorganic nutrients because the culture time is too short.

Response: We thank the reviewer for constructive comments. The dilution effect had already been corrected in the initial chl-a concentration in our original manuscript. As chl-a concentrations of coastal waters were much higher than the offshore waters in the NSCS during our cruises, the bottle volume of 2.4L could already allow us to get enough chl-a samples. For stations near the outer shelf, we have parallel experiments to make sure we have enough water for chl-a sampling. We have clarified these in the revised manuscript. We agree with the reviewer that it would be better if the incubation experiments could continue longer than two days. However, we are not allowed to perform a long period of incubation due to limitation of cruise time. On the other hand, previous results over the NSCS shelf (Li et al., 2016) indicated that phytoplankton here would react fast in the first two days of incubation and then go stable. Our nutrient data also suggest that two days of incubation are long enough to evaluate phytoplankton responses to nutrient drawdown (see our revised figure 6).

3. In the manuscript, there were no parameters concerning physiological response.

Response: We agree with the reviewer that we do not have direct measurements of physiological parameters. We have replaced "physiological response" in our title by "phytoplankton response" in the revised manuscript. Nevertheless, we believe the results of nutrient addition experiments and water mixing experiments should reflect physiological change of phytoplankton to varying nutrient conditions.

4. P6, line 1-2, the descending salinity would have obvious effect on phytoplankton growth, and the paper didn't evaluate the direct effects of salinity.

Response: We agree with the reviewer about the effect of salinity on phytoplankton growth. We have discussed this properly in the revised manuscript. Coastal phytoplankton species can generally tolerate a much larger range of salinity than estuarine and oceanic species (e.g. Brand 1984). The salinity of 6.6-30.7 during the mixing experiment at the frontal zone is higher than the lethal level of ~5 for most estuarine phytoplankton species due to osmotic pressure (Kies, 1997; Floder et al., 2010). However, inter-specific differences in salinity tolerances of phytoplankton may be important for phytoplankton growth at the lower ranch of the PRE where fluctuating salinities between 0-10 were found.

5. P8 line 16 delete "of"

Response: Done.

6. The incubation site S8 was not marked in Figure 1. The hydrographic and biogeochemical properties of S8 were not mentioned too.

Response: Done.

7. In Figure 1 the white salinity lines were marked as 22 and 32, which were described as 26 and 32.

Response: Thanks for pointing out this. We have corrected it to 26.

8. In Figure 2 "A Temperature vs. Salinity diagram during May-June 2016" should be corrected as April-June 2016.

Response: Done.

Please also note the supplement to this comment:
https://www.biogeosciences-discuss.net/bg-2017-495/bg-2017-495-AC3-

supplement.pdf

**Supplement:**

[revised manuscript text omitted]

---

## Author Response (AR2)

Dear Editor,

We have revised the manuscript based on the reviewer's suggestion. Below are our responses to his comments:

1  P5 ln 13 . Change 'Refield' to 'Redfield'

Response: Done.

2.  P10 ln 12. "decreases in net growth rate". It looks to me that there is no net growth, in fact chlorophyll has been lost between day 1 and 2. I.e., net growth is negative.

Response: Done. We have changed it to "there were negative net growth rates ..."

3.  P10, ln 18-19. "The small percentage of FPW addition (12.5%) was to ensure that the intitial chlorophyll-a concentration after FPW dilution is comparable with that of the control experiment." This sentence is confusing to me still. Is it the fact that 12.5% (I assume vol/vol addition) is a small volume that is important, or that the PW water is filtered and does not add chl to the experiment that is the key?

Response: Both the small volume and the filtered PW are important, which will ensure that the initial chlorophyll-$a$ will neither be promoted by the addition nor be diluted too much by the addition. We have clarified this in the revised manuscript.

Sincerely,

Qian Li

South China Sea Institute of Oceanology

Chinese Academy of Science, Guangzhou